# THINKING IN GROUPS: PERMUTATION TESTS REVEAL NEAR-OUT-OF-DISTRIBUTION

## ABSTRACT

Deep neural networks (DNNs) have the potential to power many biomedical work-flows, but training them on truly representative, IID datasets is often infeasible. Most models instead rely on biased or incomplete data, making them prone to out-of-distribution (OoD) inputs that closely resemble in-distribution samples. Such near-OoD cases are harder to detect than standard OOD benchmarks and can cause unreliable—even catastrophic—predictions. Biomedical assays, however, offer a unique opportunity: they often generate multiple correlated measurements per specimen through biological or technical replicates. Exploiting this insight, we introduce Homogeneous OoD (HOoD), a novel OoD detection framework for correlated data. HOoD projects groups of correlated measurements through a trained model and uses permutation-based hypothesis tests to compare them with known subpopulations. Each test yields an interpretable p-value, quantifying how well a group matches a subpopulation. By aggregating these p-values, HOoD reliably identifies OoD groups. In evaluations, HOoD consistently outperforms point-wise and ensemble-based OoD detectors, demonstrating its promise for robust real-world deployment.

## 1 INTRODUCTION

Black-box deep neural networks are increasingly used in high-stakes biomedical fields like health-care, raising significant concerns about the safety, reliability, and interpretability of AI-driven diagnostic outcomes (Rudin, 2019). While deep neural networks excel at fitting complex functions, their outputs are only as reliable as the quality and scope of their training data and objective functions. If networks receive inputs outside their training scope, they may produce misleading results, potentially leading to serious misdiagnoses.

In theory, training deep neural networks on sufficiently large datasets representing the entire target population could mitigate this issue. However, acquiring such comprehensive data is often impractical in biomedical domains for several reasons. First, *biological populations continuously evolve*. Bacteria, for example, constantly mutate to form new strains (Ahloowalia et al., 2004), making it impossible to include all variants in a single dataset. As a result, neural networks trained to classify bacterial measurements (e.g., single-cell images) into clinically relevant categories – such as species (e.g., E. Coli, Acinetobacter) (Ahmad et al., 2023; Zhang et al., 2024) – may produce incorrect predictions when given inputs from a novel strain. Second, *rare subpopulations are often underrepresented in biomedical datasets*. In digital pathology, for instance, deep neural networks are used to detect and classify cancer subtypes from biopsy images (Madabhushi & Lee, 2016). Because training datasets typically lack certain rare cancer subtypes (Boyd et al., 2016), the real-world utility of classifiers trained on such limited data is also limited. Third, *certain biological processes affecting populations may be unknown during training*. In personalized regenerative medicine, for instance, stem cells from individual patients are specialized into specific cell types (e.g., blood cells, heart muscle cells, nerve cells) and implanted to treat damaged or diseased tissue (Zakrzewski et al., 2019). Deep classifiers proposed to monitor and control this differentiation process (Zhu et al., 2021; Kusumoto & Yuasa, 2019) encounter similar issues when given patient-specific phenotypes not seen during training. Lastly, *most data acquisition processes are neither perfect nor standardized*. Each instrument or round of experiments may have slightly different settings, and it is impossible to ensure that future instruments and experiments will produce data consistent with training data. Therefore,

in many biomedical applications, deep neural networks inevitably face the risk of receiving inputs outside their training scope, making their outputs unreliable in those instances.

Formally, consider a population with $K$ subpopulations (where $K$ can be infinite). A common approach is to sample $N$ data points from different sources, where each data point maps to one of the $k$ subpopulations ($k \ll K$). These data points are then divided into training, validation, and test splits. A classifier is then trained to map each data point to one of $k$ subpopulations, and its performance is evaluated on test data from the $k$ subpopulations. This setup tests the model's ability to generalize within those subpopulations, but its effectiveness is limited to those subpopulations only. Obtaining high accuracy from such evaluations can be misleading, as it does not reflect how data points outside the $k$ subpopulations will be classified. Consequently, a seemingly well-performing model could be prone to misclassification when given real-world samples from a $(k + i)$ th subpopulation, where $i \geq 1$.

The above setting is a flavor of the more general Out-of-Distribution (OoD) detection problem, and such inputs are typically called near OoD data (Fort et al., 2021). For practical applications, models must not only determine whether a test sample belongs to a known subpopulation, but also detect when it does not (Hendrycks et al., 2021). Although OoD detection is a well-studied problem (see Section 2.1), near-OoD detection remains challenging and largely unsolved (Yang et al., 2021). In particular, when a new data point lies close to the training manifold, it is hard to determine whether it is inside or outside the training distribution. Conventional OoD detection methods struggle in this regime, often yielding unreliable, often catastrophic, predictions.

An often overlooked aspect in many biomedical applications is that one specimen often contains several replicates (e.g., several bacteria cells in a clinical sample, several image patches in a biopsy slide, and several stem cells in a regenerative medicine). **During inference, one specimen yields multiple data points that, by definition, arise from the same subpopulation.** These data points exhibit similarities (e.g., subject characteristics, lighting conditions, and camera settings), making them inherently correlated. While this is a serious limitation to generate training data, it offers a unique advantage for OoD detection. In particular, if test-time inputs arrive as a "homogeneous" sample of data points, we can frame OoD detection as a two-sample hypothesis test and operate on sample-level OoD statistics rather than aggregating noisy, point-wise OoD scores. Exploiting distributional information about the sample in this manner makes near-OoD detection more feasible. While similar concepts have been used to improve model predictions (Ahmad et al., 2023; Zieliński et al., 2017), to the best of our knowledge, it has not been used for OoD detection. In this work, we exploit this concept to formulate a novel OoD detection method, which we term homogeneous-OoD detection or "HOoD".

**Definition 1** (Homogeneity). *Given a sample $\{x_i\}_{i=1}^{N} \subset X$ and subpopulations $\mathcal{Y} = \{y_1, y_2, \ldots, y_K\}$, the sample is homogeneous w.r.t. subpopulation $y_k$ if a mapping $f : X \to \mathcal{Y}$ exists and $f(x_i) = y_k \ \forall i = 1, \ldots, N$*

With this definition, we reformulate OoD detection as a hypothesis test between homogeneous sets. Hypothesis testing is a fundamental statistical concept that involves making assumptions about populations and rigorously evaluating whether the sample data provide sufficient evidence to support or reject these assumptions. At test time, we assume we are given a homogeneous sample of data points and need to classify the entire sample as either InD or OoD. Specifically, we test the exchangeability of this sample with representative samples from each of $k$ subpopulations, through permutation testing on latent responses. Our method is independent of the choice of latent response, allowing to choose them based on model architecture. We evaluate our approach across several models, datasets, latent responses, and label splits, demonstrating that computing statistical significance enables effective OoD detection of homogeneous samples, with the null distribution providing interpretability. Our key contributions are as follows:

- We identify a previously overlooked OoD detection setting where inputs are homogeneous samples from a single subpopulation, often seen in biomedical applications like healthcare and diagnostics.

- We define a new OoD detection task tailored to this setting, and show that standard point-wise OoD detection methods generate mixed results under this setting.

- We introduce a new OoD detection method for homogeneous samples based on two-sample hypothesis testing and permutation tests, overcoming the limitations of point-wise OoD detection methods.

## 2 BACKGROUND

### 2.1 OUT OF DISTRIBUTION DETECTION

OoD detection is the process of detecting inputs that fall outside a model's training data distribution (Yang et al., 2021). Depending on what is assumed in-distribution (InD), an OoD data point can be categorized as near-OoD or far-OoD. Near-OoD data points are semantically similar to, but outside, the training distribution. For example, if a classification model $M$ is trained only on handwritten MNIST digits 0–4 (InD), then MNIST digits 5–9 are near-OoD to $M$. If we instead consider a similar digit dataset, SVHN (Netzer et al., 2011), its digits 0–4 are near-OoD to $M$ (semantically similar, yet outside the training distribution), while its digits 5–9 are far-OoD to $M$. Note that in order to reflect real-world applications, our setting requires stricter boundaries than most OOD benchmarks. For instance OpenOOD benchmark (Yang et al., 2022) defines more relaxed boundaries between near- and far-OoD; for example, with MNIST as InD, NotMNIST and Fashion-MNIST are categorized as near-OoD, while Texture, CIFAR-10, TinyImageNet, and Places365 are categorized as far-OoD.

Near-OoD detection is more challenging than far-OoD detection because, despite being OoD, near-OoD data is semantically similar to training data (Fort et al., 2021). (See Appendix A.6 for an example) The effectiveness of OoD detection can vary across models, datasets, domains, and definitions of InD (Yang et al., 2021). Recent work shows that, without a precise definition of InD, any single-sample OoD test can be fooled—there always exists a setting where single-sample tests perform no better than chance (Zhang et al., 2021). Given the limitations of single-sample OoD tests, we identify that OoD detection can be approached via point-wise methods, which evaluate individual data points, or batch-wise methods, which use collective statistics of data batches, with formal definitions provided in Appendix A.1, A.2.

### 2.2 RELATED WORK ON POINT-WISE OOD DETECTION

Recent years have seen extensive work on point-wise OoD detection. Several works propose score functions $\mathcal{X} : \mathbb{R}^D \rightarrow \mathbb{R}$ to quantify the OoD-ness of a data point relative to a pre-trained model (Hendrycks & Gimpel, 2016; Liu et al., 2020; Sun et al., 2022). Common score functions include maximum softmax probability (MSP) (Hendrycks & Gimpel, 2016), which takes the highest softmax output; the energy score (Liu et al., 2020), derived from classifier logits; and Deep Nearest Neighbors (Sun et al., 2022), which takes the mean nearest-neighbor distance in latent space. Likelihood-based methods have also been explored; however, deep generative models often assign higher likelihoods to OoD data than InD data (Nalisnick et al., 2019), making likelihood an unreliable metric. For example, a flow-based model trained on CIFAR-10 frequently assigns higher likelihood to SVHN images, despite clear visual differences. Statistical hypothesis tests for OoD detection, such as Goodness-of-Fit tests on typical sets (Nalisnick et al., 2019) and Likelihood Ratio tests (Ren et al., 2019), have also been proposed. In addition, OoD-aware training methods – such as Outlier Exposure (Hendrycks et al., 2018), OECC (Papadopoulos et al., 2021), and Evidential Learning (Sensoy et al., 2018) – improves OoD detection performance during training, while post-hoc calibration methods like ODIN (Liang et al., 2017), improves OoD detection performance after training via input preprocessing and temperature scaling. Unlike above methods, our method frames OoD detection as a two-sample hypothesis test with a homogeneity prior on test samples, and computes statistical significance using permutation tests rather than distributional assumptions.

### 2.3 RELATED WORK ON BATCH-WISE OOD DETECTION

Several methods have explored batch- or group-wise OoD detection, though with different assumptions and techniques than HOoD. DisCoPatch (Caetano et al., 2025) utilizes batch normalization (BN) statistics (mean/variance) to detect semantic, domain, and covariate shifts by dividing images into patches and treating them as a batch. Similarly, Song et al. (2019) use BN in generative models

(VAEs, PixelCNN) for unsupervised detection, switching to training mode at test time to recompute batch statistics and derive a permutation-based "drop score." Bergamin et al. (2022) uses Fisher information to aggregate log-likelihoods and gradients from generative models into batch p-values. Jiang et al. (2021) use flow-based models with random projections and Kolmogorov–Smirnov tests for group scoring. DynaSubVAE (Behrouzi et al., 2025) clusters VAE latents into subgroups and uses a "regret" score for OoD, incorporating group structure but still generative-model-dependent. Other approaches modify training rather than test-time detection: Weighted Non-IID Batching (WNB) (Zhao & Cao, 2024) reweights batches to improve standard OoD detectors, and Boundary Aware Learning (Pei et al., 2022) employs batch-wise training but point-wise detection. In contrast, HOoD offers a model-agnostic, test-time batch-wise framework that explicitly tests exchangeability.

## 2.4 Statistical Foundations for HOoD

We build on two strands of statistical methodology: (i) Permutation Tests, which assess the exchangeability of unseen batches with reference batches without strong distributional assumptions, and naturally extend to the batch-OoD setting, and (ii) Conformal Prediction (CP) (Shafer & Vovk, 2008), which uses calibration sets and non-conformity scores to provide finite-sample guarantees. We describe Conformal Prediction in the Appendix A.3. The rest of this section briefly introduces Permutation tests. Permutation tests are a type of nonparametric hypothesis test, frequently used when the assumptions of traditional parametric tests (e.g., normality, homoscedasticity) are not met(Good, 2013; Tibshirani & Efron, 1993). A two-sample permutation test is conducted as follows: let $T$ be a test statistic, such as the difference of means. Given two samples $X_A = x_1, \cdots, x_n$ and $X_B = x_{n+1}, \cdots, x_N$ with $1 \leq n < N$, we test the null hypothesis $H_0$ that $X_A$ and $X_B$ are exchangeable. First, compute the observed statistic $T_{obs}$. Then enumerate the $t = \frac{N!}{n!(N-n)!}$ possible assignments of $X_A \cup X_B$ into two sets of size $n$ and $(N-n)$, and compute $T_1, \cdots, T_t$. Finally, calculate the proportion of arrangements with $T_t > T_{obs}$ (one-tailed) or $|T_t| > |T_{obs}|$ (two-tailed) to obtain the p-value for $H_0$. Because $t$ grows factorially with $N$, Monte Carlo sampling methods are often used to approximate the null distribution, and advances in computational power and Monte Carlo techniques (Good, 2013) have enabled their application to large, high-dimensional datasets. In our framework, the notion of using reference sets for comparison is borrowed from CP, while permutation tests provide the mechanism for evaluating whether batches are homogeneous and exchangeable. Together, these two strands form the statistical foundation of our approach.

## 3 Methodology

Fig. 1 shows an illustration of the overall procedure, from receiving a homogeneous sample to obtaining the OoD decision. Let $D_T = \{x_i, y_i\}_{i=1}^{N_T}$ be training data, $D_V = \{x_i, y_i\}_{i=1}^{N_V}$ be validation data, $D_I = \{x_i, y_i\}_{i=1}^{N_I}$ be InD test data, $D_O = \{x_i, y_i\}_{i=1}^{N_O}$ be OoD test data, and $\omega = \{\omega_1, \ldots, \omega_K\}$ be the label space of $D_T$ (same for $D_V$). Here, $x \in \mathbb{R}^D$ and $y \in \omega$. We quantify how a model $\phi$ trained on $D_T$ and validated on $D_V$ would transform homogeneous samples from $D_I$ and $D_O$, using latent responses $Z(x_i; \phi) : \mathbb{R}^D \to \mathbb{R}^L$ where $Z = [Z_1, \ldots, Z_L]$. We leave the choice of latent responses $Z$ for future investigation and instead use well-established methods from the literature. We then define $K$ null hypotheses to test exchangeability.

$$\mathbf{H_0^k} : \textbf{The new sample is exchangeable with group } \boldsymbol{\omega_k}$$

For each $H_0^k$, we first compute the MRPP statistic $\delta(\pi_k^{obs})$ for the true assignment $\pi_k^{obs}$. The formal definition of MRPP is provided in Appendix A.4. Second, we perform $P$ randomized assignments $\pi_k^i$ of the union of the test sample and the $k$-th sample, and compute the MRPP statistic $\delta(\pi_k^i)$ under each assignment. Third, we compute the proportion $p_k$ of assignments that yield $\delta(\pi_k^i) \leq \delta(\pi_k^{obs})$. Fourth, we store $\{\delta(\pi_k^{obs}), p_k\}$ as the outcome of $H_0^k$. We then repeat this procedure for all $H_0^k$, and compute two vectors, $\delta = [\delta(\pi_1^{obs}), \cdots, \delta(\pi_k^{obs})]$ and $\mathbf{p} = [p_1, \cdots, p_k]$. Finally, we use $\mathbf{p}$ to accept or reject the composite hypothesis that the test sample is exchangeable with one of $k$ samples, under a significance level $\alpha$.

$$D_\alpha(x) = \begin{cases} \text{InD} & ; \text{if } \max_k\{p_k\} \geq \alpha \\ \text{OoD} & ; \text{otherwise} \end{cases}$$

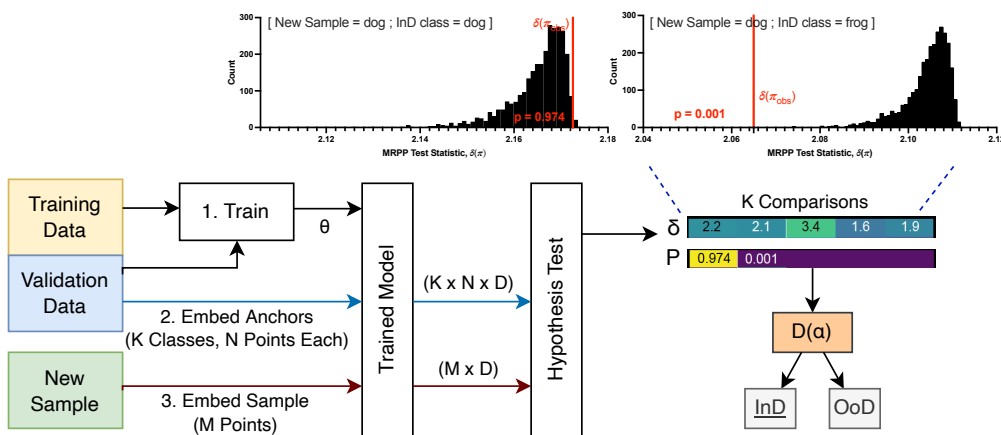

Figure 1: Method Overview: A dataset with training and validation data from classes $\{y_1, \cdots, y_k\}$ with $(k \ll K)$ is used to train a model. Next, for each class $y_k$, a homogeneous sample $S[y_k]$ ($N$ data points) is drawn from validation data and passed through the model to obtain latent responses. At inference time, a new homogeneous sample ($M$ data points) is passed through the model to obtain latent responses, and compared with each $S[y_k]$ using hypothesis tests. Each test yields a statistic $\delta_k$ and significance value $p_k$, which is then vectorized ($\delta$, $\mathbf{p}$). The distribution generated for each of $k$ comparisons provide insight to the homogeneity and exchangeability of the test sample. Finally $\mathbf{p}$ is mapped into InD/OoD using a decision function $D(\alpha)$, at significance level $\alpha$.

## 4  EXPERIMENTS

We conduct several experiments to validate the effectiveness of the proposed method. Additional experiments and results can be found in the Appendix A.6, A.7.

### 4.1  EXPERIMENT SETUP

We first setup a toy near-OoD detection task using the benchmark MNIST and CIFAR10 datasets to validate our method. We then explore a real homogeneous-OoD detection task with a labeled dataset, **AMRB** Ahmad et al. (2023), containing single-cell bacteria images. To span common model architectures, we train classifiers (ResNet-50, ResNet-18, ResNet-EDL), auto-encoders (ResNet-AE), and classifier-autoencoder hybrids (ResNet-CAE) on different subsets of the data. All models were implemented in PyTorch Lightning, and trained on NVIDIA A100 GPUs. For the ResNet-50 and ResNet-18 models, we use the standard, ImageNet pre-trained implementation provided in PyTorch and use cross-entropy loss for training. For the ResNet-AE models, we use a convolutional encoder $F$ and decoder $G$ with residual connections, and use reconstruction (MSE) loss for training. For the ResNet-CAE models, we add a classification head $C$ onto the bottleneck layer of ResNet-AE, and use a joint cross-entropy and weighted (0.5) reconstruction (MSE) loss for training. For the ResNet-EDL models, we use the same structure as ResNet-CAE, but use evidential loss (Sensoy et al., 2018) for training.

The model parameters were optimized for 100 epochs, using Adam with a learning rate of 0.001. Upon training, we observe that ResNet-50 models yield a high accuracy across all cases. The classification performances are reported in Appendix A.5. To cover a range of training procedures and models, we choose the ResNet-50, ResNet-CAE, and ResNet-AE models to assess our method.

### 4.2  TOY PROBLEM: MNIST AND CIFAR10

Here, we define a toy homogeneous-OoD detection problem to assess our method, using subsets from **MNIST** and **CIFAR10**. Both datasets contain samples from 10 different classes. We first split them into two subsets (A and B) based on class label. For MNIST, we use $A = \{0, 1, 2, 3, 4\}$ and $B = \{5, 6, 7, 8, 9\}$. For CIFAR-10, we use $A = \{\text{dog, frog, horse, ship, truck}\}$ and $B = \{\text{plane, automobile, bird, cat, deer}\}$. Here, InD samples of A are near-OoD for B, and vice versa.

We train separate classifiers for labels sets A and B, and apply our method to compare the 10 test set classes, against the $L$ validation set classes of each split. We expect to see low p-values when comparing samples from different classes, and high p-values when comparing samples from the same class. Table. 1 shows results from passing IND and OOD homogeneous test samples to different splits of models. Most off-diagonal comparisons yielded $p < 0.05$, demonstrating that our method can detect non-interchangeable samples, and consequently, OoD samples (i.e. samples that are non-interchangeable with any reference sample).

Table 1: Evaluation of HOoD predictions across Split A and Split B of MNIST and CIFAR-10 datasets. P-values are obtained from HOoD tests using logit outputs of four models trained on disjoint subsets of MNIST and CIFAR-10 labels. Each row represents the ground truth OOD decision, p-value and the predicted OOD decision for a single test sample (Statistic = MRPP, Permutations = 3000, Sample Size = 100). Red color indicates incorrect predictions.

| Dataset | Test Sample | Split A | | | Split B | | |
| | | Ground Truth | max $p$ | Prediction $D(\alpha = 0.001)$ | Ground Truth | max $p$ | Prediction $D(\alpha = 0.001)$ |
|---|---|---|---|---|---|---|---|
| **MNIST** | 5 | IND | 0.765 | IND | OOD | $< \alpha$ | OOD |
| | 6 | IND | 0.803 | IND | OOD | $< \alpha$ | OOD |
| | 7 | IND | 0.441 | IND | OOD | $< \alpha$ | OOD |
| | 8 | IND | 0.458 | IND | OOD | $< \alpha$ | OOD |
| | 9 | IND | 0.710 | IND | OOD | $< \alpha$ | OOD |
| | 0 | OOD | $< \alpha$ | OOD | IND | 0.551 | IND |
| | 1 | OOD | $< \alpha$ | OOD | IND | 0.128 | IND |
| | 2 | OOD | $< \alpha$ | OOD | IND | 0.042 | IND |
| | 3 | OOD | $< \alpha$ | OOD | IND | 0.166 | IND |
| | 4 | OOD | $< \alpha$ | OOD | IND | 0.609 | IND |
| **CIFAR-10** | dog | IND | 0.974 | IND | OOD | 0.108 | IND |
| | frog | IND | 0.634 | IND | OOD | $< \alpha$ | OOD |
| | horse | IND | 0.623 | IND | OOD | $< \alpha$ | OOD |
| | ship | IND | 0.391 | IND | OOD | 0.001 | IND |
| | truck | IND | 0.436 | IND | OOD | 0.002 | IND |
| | plane | OOD | $< \alpha$ | OOD | IND | 0.951 | IND |
| | automobile | OOD | $< \alpha$ | OOD | IND | 0.349 | IND |
| | bird | OOD | $< \alpha$ | OOD | IND | 0.423 | IND |
| | cat | OOD | $< \alpha$ | OOD | IND | 0.545 | IND |
| | deer | OOD | $< \alpha$ | OOD | IND | 0.965 | IND |

## 4.3 DOMAIN PROBLEM: AMRB

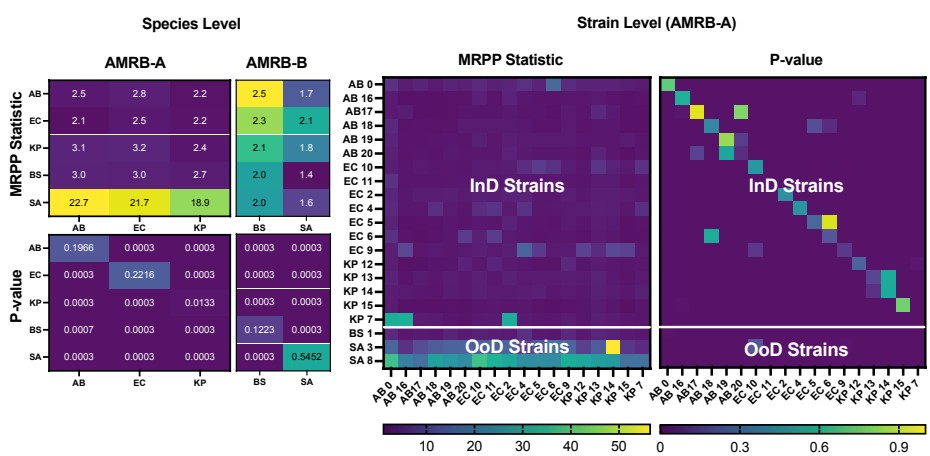

Figure 2: MRPP statistic and P-value from $5 \times 5$ species-level HOoD tests (Left) and $21 \times 18$ strain-level HOoD tests (Right) performed using logit outputs from 2 ResNet-50 models (AMRB-A, AMRB-B) each trained on disjoint subsets of labels from the AMRB dataset. Each cell represents a HOoD test (Permutations = 3000, Sample Size = 100) between a test sample (row-label) and a reference sample (column-label). High $p$-values expected for HOoD tests on the diagonal (same species/strain).

Here, we explore a real homogeneous-OoD detection problem with a labeled dataset, **AMRB** (Ahmad et al., 2023), containing singlecell bacteria images taken from 21 strains across 5 *species* $\{Ab, Bs, Ec, Kp.Sa\}$. Table 2 provides a breakdown of the homogeneous groups within the AMRB dataset.

Table 2: Homogeneous Groups within the AMRB Dataset: **W:** wild, **N:** non-wild

| $\sum$ | Species | Gram | Morphology | $\sum_W$ | $\sum_N$ |
|---|---|---|---|---|---|
| 5 | Ab | $-$ | Rod | 0 | 5 |
| 2 | Bs | $+$ | Rod | 2 | 0 |
| 7 | Ec | $-$ | Rod | 2 | 5 |
| 5 | Kp | $-$ | Rod | 0 | 5 |
| 2 | Sa | $+$ | Spherical | 1 | 1 |
| 21 | 5 | 2 | 2 | 5 | 16 |

From this dataset, we take 2 subsets based on species, $A = \{Ab, Ec, Kp\}$ and $B = \{Bs, Sa\}$. We then train ResNet-50, ResNet-18, and ResNet-CAE models on each split, and assess their OOD detection performance under two homogeneity levels, (a) species ($K \in \{3, 2\}$) and (b) strain ($K \in \{18, 3\}$). The classification performances of each model and split are reported in Appendix A.5. Fig. 2 reports our findings for species-homogeneous tests (left) and strain-homogeneous tests (right).

In the species-homogeneous case, the on-diagonal (same-species) tests showed higher p-values than the off-diagonal (different-species) tests. The strain-homogeneous case showed a similar trend, with on-diagonal (same-strain) tests showing higher p-values than the off-diagonal (different-strain) tests. However, certain off-diagonal (different-strain) tests showed higher p-values than the others. Upon closer examination, we find that most off-diagonal (different-strain) tests with higher p-values involve strains that share the same species. This species-level similarity most likely contributes to their higher p-values.

### 4.4 COMPARISON WITH EXISTING METHODS

We evaluate our approach on the task of labeling an entire batch of homogeneous samples as In-Distribution (InD) or Out-of-Distribution (OoD). We use three types of baselines in our evaluation. The first type is aggregated *point-wise OoD scores* (MSP). Here, given an input sample, we compute an OoD score per data point, and reduce them to a scalar by taking mean/max. The second type is aggregated *point-wise OoD scores that use calibration data* (CP, CPP). Here, given an input sample, we compute $K$ OoD scores per data point (one per reference set), and reduce them to a $K$-dimensional vector by taking mean/max. The third type (ours) is *two-sample permutation tests* (MRPP, LSP, MD). Here, given an input sample, we perform $K$ two-sample tests (one per reference set) using permutations, and obtain a $K$-dimensional vector of $p$-values.

We apply the above methods on 100 class-conditional random samples from the test set, and compute an OoD score/vector per sample. Next, we convert each score/vector into a binary InD/OoD decision, and assess the performance of this decision-making process using the AUC metric. The AUC is suitable for this comparison because it provides a robust, threshold-independent measure of the model's ability to distinguish between InD and OoD samples, effectively capturing the trade-off between true positive and false positive rates (Fawcett, 2006). We repeat this process for each method and class.

The specifics of each method are described below: **MRPP** (Mielke & Berry, 2007): Permutation tests with *MRPP* as test statistic for homogeneity, **LSP** (Humblot-Renaux et al., 2023): Permutation tests with *AUC* as test statistic for linear separability, **MD** (Kim, 2015): Permutation tests with *Mean Distance* as test statistic for the average difference, **MSP** (Hendrycks & Gimpel, 2016): average of prediction *classes*, using Maximum Softmax Probability, **CP** (Shafer & Vovk, 2008): average of predicted *sets* using vanilla CP (works only on logit spaces), and **CPP** (Shafer & Vovk, 2008): average of predicted *sets*, using a modified CP with a kernel function $F : \mathbb{R}^d \rightarrow \mathbb{R}^k$ (centroid distance) to run CP beyond logits.

Table 3 reports the OoD detection AUC when using logit-space latent responses of models ResNet-EDL, ResNet-CAE, and ResNet-50, respectively. These models were evaluated using six OoD detectors: MRPP, LSP, MD, MSP, CP, and CPP. Here, MRPP and LSP, which use permutation

Table 3: Comparison of OoD Detection with logits as latent responses, using AUC as the performance metric (higher is better). Results computed over 50 InD/OoD classification runs comparing random test samples against validation set anchors. † indicates the use of permutation tests.

| | ResNet-EDL | | ResNet-CAE | | ResNet-50 | |
|---|---|---|---|---|---|---|
| **Stat** | A | B | A | B | A | B |
| **MNIST** | | | | | | |
| MRPP† | **1.00** | .998 | **1.00** | **1.00** | **1.00** | .998 |
| LSP† | .998 | **1.00** | **1.00** | **1.00** | **1.00** | .994 |
| MD† | .997 | .975 | .885 | .762 | .877 | **.998** |
| MSP | .657 | .389 | .638 | .725 | .682 | .719 |
| CP | **1.00** | .971 | **1.00** | **1.00** | **1.00** | **1.00** |
| CPP | .913 | .840 | **1.00** | **1.00** | **1.00** | .844 |
| **CIFAR-10** | | | | | | |
| MRPP† | .979 | **1.00** | .987 | **1.00** | **1.00** | **1.00** |
| LSP† | .977 | **1.00** | .978 | **1.00** | **1.00** | **1.00** |
| MD† | .666 | .900 | .800 | .890 | .928 | .967 |
| MSP | .726 | .440 | .751 | .449 | .693 | .587 |
| CP | .701 | .762 | .673 | .758 | .801 | **.995** |
| CPP | .757 | .726 | .726 | .755 | .785 | **.998** |
| **AMRB-SP** | | | | | | |
| MRPP† | **1.00** | .983 | **1.00** | .969 | .996 | .995 |
| LSP† | .999 | .961 | **1.00** | .954 | .991 | **1.00** |
| MD† | .898 | .959 | .862 | .716 | .560 | **.993** |
| MSP | .707 | .500 | .736 | .500 | .748 | .500 |
| CP | .886 | **1.00** | .880 | **1.00** | .804 | **1.00** |
| CPP | .871 | **.999** | .879 | **1.00** | .853 | **1.00** |

tests, performs the highest, especially for MNIST and AMRB-SP datasets. In contrast, the baselines like CPP, CP, and MSP showed lower performance, especially on CIFAR10, with CPP and MSP underperforming in most cases.

Table 4: Comparison of OoD Detection methods with OoD metrics as latent responses, using AUC as the performance metric (higher is better). Results computed over 50 InD/OoD classification runs comparing random test samples against validation set anchors. † indicates the use of permutation tests.

| | ResNet-AE | | ResNet-EDL | | ResNet-CAE | | ResNet-50 | |
|---|---|---|---|---|---|---|---|---|
| **Stat** | A | B | A | B | A | B | A | B |
| **MNIST** | | | | | | | | |
| MRPP† | .992 | .993 | **1.00** | .998 | **1.00** | .998 | **1.00** | .998 |
| LSP† | 1.00 | .982 | **1.00** | .998 | **1.00** | **1.00** | **1.00** | **1.00** |
| MD† | .745 | .963 | .929 | .774 | **1.00** | .906 | .868 | .987 |
| CPP | .810 | .797 | .801 | .503 | .808 | .658 | .819 | .798 |
| **CIFAR-10** | | | | | | | | |
| MRPP† | .891 | .858 | .964 | **1.00** | .960 | **1.00** | .999 | **1.00** |
| LSP† | .865 | .773 | .954 | **.998** | .945 | **.999** | .992 | **1.00** |
| MD† | .686 | .594 | .742 | .734 | .837 | .753 | .638 | .837 |
| CPP | .556 | .574 | .500 | .500 | .500 | .500 | .500 | .500 |
| **AMRB-SP** | | | | | | | | |
| MRPP† | .996 | .983 | **1.00** | .998 | **1.00** | .994 | .992 | .995 |
| LSP† | 1.00 | .995 | .999 | .985 | **1.00** | .998 | .945 | **1.00** |
| MD† | .941 | .683 | .885 | .712 | .930 | .710 | .875 | .591 |
| CPP | .580 | **.973** | .500 | .500 | .639 | .500 | .500 | .500 |

Table 4 reports the OoD detection AUC when using OoD-metric latent responses of models ResNet-EDL, ResNet-CAE, ResNet-50, and an auto-encoder ResNet-AE, respectively. These models were evaluated using four OoD detectors: MRPP, LSP, MD, and CPP. Similar to Table 3, MRPP and

LSP perform the best, achieving near-perfect detection across all datasets. ResNet-AE with MRPP achieves particularly high accuracy on MNIST and AMRB-SP, while LSP demonstrates the best results for CIFAR10. In contrast, methods such as CPP and MD generally underperform. Using permutation tests (MRPP, LSP) significantly enhanced the performance of the models in OoD detection tasks.

Despite using permutation tests, MD underperforms relative to MRPP and LSP across the datasets. MD relies on the difference between sample means, which implicitly assumes that the data follows a unimodal distribution. This assumption often does not hold in real-world data, leading to suboptimal performance. Consequently, MD proves less effective as an OoD indicator in these cases.

## 5 DISCUSSION AND CONCLUSION

We identified an overlooked setting that is common in biomedical applications, where inputs are homogeneous samples from a subpopulation. We formulated a new OoD detection problem for this setting, homogeneous-OoD, and introduced a novel method to detect homogeneous-OoD. Given a trained model, $K$ homogeneous reference samples from InD subpopulations, and an unknown sample, we use two-sample hypothesis testing and permutation tests to assess its exchangeability with each of $K$ samples. We then apply the MRPP statistic on model-generated latent responses of each data point in the samples, and derive the p-value of the statistic through permutation tests.

To validate our method, we trained classifiers on a subset of classes from the MNIST and CIFAR-10 datasets, and evaluated OoD performance using samples from excluded classes. We applied our method on a real homogeneous-OoD detection task, using a bacteria classifier trained on few species, and demonstrated a practical use of our method on detecting unseen species. Results suggest that this approach can detect samples in the near-OoD regime, addressing a key challenge in real-world OoD detection.

Our method is model-agnostic and imposes no restriction on the type of model or latent responses used. However, as shown in our experiments, OoD detection performance may vary depending on model architecture, training procedure, latent response type, and the chosen test statistic. Although we employed the MRPP statistic in our analysis, The proposed procedure is compatible with any well-defined test statistic of form $\mathbb{R}^d \rightarrow \mathbb{R}$, to capture other types of sample differences. The emphasis on homogeneity provided by MRPP yielded consistent performance across a variety of datasets and models. In the future, we plan to refine this framework and explore additional domains where homogeneous-OoD detection is critical to performance and safety. Overall, our method, focusing on homogeneity and offering the flexibility to integrate alternative test statistics, provides a foundation for reliable testing of classifiers in preparation for real-world deployment.

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

# A APPENDIX

## A.1 POINT-WISE OoD DETECTION

Let,

$$D_T = \{(x_i, y_i)\}_{i=1}^{N_T}, \quad D_V = \{(x_i, y_i)\}_{i=1}^{N_V}, \quad D_I = \{(x_i, y_i)\}_{i=1}^{N_I}, \quad D_O = \{(x_i, y_i)\}_{i=1}^{N_O}$$

be the training, validation, in-distribution test, and out-of-distribution test datasets, respectively, with label space

$$\omega = \{\omega_1, \ldots, \omega_K\}, \quad y \in \omega.$$

Let $\phi$ be a model trained on $D_T$ and validated on $D_V$, and define the latent response mapping

$$Z(x; \phi) : \mathbb{R}^D \to \mathbb{R}^L, \quad Z(x; \phi) = [Z_1(x), \ldots, Z_L(x)].$$

A *point-wise OoD detector* evaluates each input $x \in \mathbb{R}^D$ independently by computing an OoD score

$$S(x) = g(Z(x; \phi)), \quad S(x) \in \mathbb{R},$$

where $g(\cdot)$ is a scoring function (e.g., maximum softmax probability, energy score, Mahalanobis distance). The decision rule is

$$x \in \text{OoD} \quad \Longleftrightarrow \quad S(x) > \tau,$$

for some threshold $\tau$ calibrated on $D_V$. Under the in-distribution hypothesis $H_0$, the joint likelihood factorizes as

$$P(x_1, \ldots, x_n \mid H_0) = \prod_{i=1}^{n} P(x_i \mid H_0),$$

so each test point can be judged *independently of the others*.

## A.2 BATCH-WISE OoD DETECTION

Using the same notation as Appendix A.1, consider a batch of related test points

$$B = \{x_1, \ldots, x_n\}, \quad x_i \in \mathbb{R}^D,$$

where all points in $B$ share the same attributes (e.g., drawn from the same environment, domain, or time window). A *batch-wise OoD detector* evaluates $B$ collectively by posing the hypothesis test

$$H_0 : B \sim P_{\text{InD}}, \quad H_1 : B \not\sim P_{\text{InD}}.$$

The method computes an *aggregate test statistic*

$$T(B) = f\big(\{Z(x_i; \phi)\}_{i=1}^{n}\big),$$

where $f(\cdot)$ summarizes the joint latent responses of the batch. The decision rule is,

$$B \in \text{OoD} \quad \Longleftrightarrow \quad T(B) \text{ exceeds the } (1-\alpha)\text{-quantile of the null distribution of } T(B) \text{ under } H_0,$$

where $\alpha \in (0, 1)$ is the chosen significance level, i.e., the maximum tolerable probability of incorrectly rejecting the null hypothesis when it is true. The null distribution may be obtained analytically (under parametric assumptions) or estimated empirically from in-distribution reference batches (e.g., via permutation or bootstrap). Unlike the in-distribution case, where independence implies factorization,

$$P(x_1, \ldots, x_n \mid H_0) = \prod_{i=1}^{n} P(x_i \mid H_0),$$

under the out-of-distribution hypothesis, the joint distribution generally does not factorize,

$$P(x_1, \ldots, x_n \mid H_1) \neq \prod_{i=1}^{n} P(x_i \mid H_1),$$

so the test must rely on the *collective distributional behavior of the batch* rather than independent point scores.

## A.3 Conformal Prediction (CP)

Conformal Prediction (CP) (Shafer & Vovk, 2008) is a statistical framework for constructing prediction sets (or intervals) with guaranteed finite-sample coverage, without strong distributional assumptions about the data (Angelopoulos & Bates, 2021). Given a trained model $M$, a calibration set $D=\{x_i, y_i\}_{i=1}^{n}$, a non-conformity function $S : \mathcal{X} \times \mathcal{Y} \to \mathbb{R}$, and a desired significance level $\alpha$, CP finds a prediction set (or region) $C_\alpha(x) = \{y : S(x, y) \leq Q(1-\alpha)\}$, where $Q(1-\alpha)$ is the $\lceil \frac{(n+1)(1-\alpha)}{n} \rceil$-th quantile of $\{S(x_i, y_i)\} \ \forall D$. The $C_\alpha(x)$ thus obtained contains the true outcome with a probability of at least $(1-\alpha)$.

Unlike traditional approaches, CP adapts to the complexity of the problem at hand, offering prediction sets that can vary in size based on the difficulty of each individual prediction. The nonconformity score in CP can serve as an indicator for OoD, with high nonconformity scores indicating higher uncertainty, and possibly OoD (Novello et al., 2024). Recent advancements have extended CP to handle challenges such as covariate shift (Tibshirani et al., 2019), online learning scenarios (Angelopoulos et al., 2024), and applications in high-dimensional spaces (Ding et al., 2024). However, CP is inherently point-wise and overlooks the group-level distributional structure our method leverages. We borrow the idea of calibration sets from CP to build reference distributions for hypothesis testing. Unlike the point-wise ranking used in CP, we compute OoD significance through permutation-based statistics.

## A.4 Multi-Response Permutation Procedure (MRPP)

MRPP (Mielke & Berry, 2007) is a permutation test for exchangeability of data points between two or more groups. Let $\boldsymbol{\Omega} = \{\Omega_1, \ldots, \Omega_N\}$ be a set of $\mathbb{R}^D$ samples. Let $\omega = \{\omega_1, \ldots, \omega_K\}$ be a set of groups with sizes $\{N_1, \ldots, N_K\}$ and $\sum_{k=1}^{K} N_k = N$. Let $\Delta : \Omega \times \Omega \to \mathbb{R}$ be a measure of dissimilarity between two samples, such as Euclidean distance. Given an assignment $\pi : \Omega \to \omega$ of samples to groups and an indicator $\Psi_k(\Omega_I; \pi) = \mathbf{1}[\pi(\Omega_I) = \omega_k]$ that denotes whether a data point $\Omega_I$ belongs to group $\omega_k$ under the assignment $\pi$, MRPP computes a test statistic $\delta$ for the mean dissimilarity within the group of samples under that assignment:

$$\delta(\pi) = \sum_{k=1}^{K} C_k \, \xi_k(\pi)$$

$$\xi_k(\pi) = \frac{\sum_{1 \leq I < J \leq N} \Delta(\Omega_I, \Omega_J) \, \Psi_k(\Omega_I; \pi) \, \Psi_k(\Omega_J; \pi)}{{}_{N_k}C_2}$$

Here, $C_k$ is a weight (e.g., $1/N_k$) that balances the contribution of each group to $\delta(\pi)$. Next, $\delta(\pi)$ is calculated across all possible assignments $\pi$, and the proportion of assignments with $\delta(\pi)$ as extreme as $\delta(\pi_{obs})$ is reported, where $\pi_{obs}$ is the observed assignment. For this particular statistic, if $\pi_{obs}$ is significant, we expect $\delta(\pi_{obs})$ to be lower than most $\delta(\pi)$, which, in turn, would place $\delta(\pi_{obs})$ at the left tail of the distribution of $\delta(\pi)$. On the other hand, if $\pi_{obs}$ is not significant, we expect $\delta(\pi_{obs})$ to be in the typical region of $\delta(\pi_i)$ values.

## A.5 ACCURACY OF TRAINED CLASSIFIERS

To ensure a fair comparison, we also report the standard test accuracies of the models on the MNIST, CIFAR-10, and AMRB datasets, which reflect their in-distribution performance prior to evaluating OoD detection (Table 5).

Table 5: Test Accuracy of Models used for OoD detection, for MNIST, CIFAR10, and AMRB datasets.

| Dataset | ResNet-50 | ResNet-18 | ResNet-CAE |
|---|---|---|---|
| **MNIST All** | 0.994 | **0.995** | **0.995** |
| **MNIST A** | 0.997 | 0.997 | 0.996 |
| **MNIST B** | **0.999** | **0.999** | 0.998 |
| **CIFAR10 All** | **0.883** | 0.815 | 0.817 |
| **CIFAR10 A** | **0.958** | 0.925 | 0.917 |
| **CIFAR10 B** | **0.929** | 0.906 | 0.856 |
| **AMRB All** | **0.761** | 0.747 | 0.735 |
| **AMRB A** | 0.993 | **0.994** | 0.993 |
| **AMRB B** | **0.787** | 0.773 | 0.766 |

## A.6 InD/OoD INSEPARABILITY AT POINT LEVEL

Fig. 3 visualizes the feature and logit spaces (2D UMAP) of three ResNet-50 classifiers trained on a single-cell bacteria dataset (AMRB) with common loss functions. One classifier was trained on all classes, while the other two classifiers were trained on disjoint subsets of classes. The classifier trained on all classes projected data points onto distinct class-wise regions, in both feature and logit spaces. However, both classifiers trained on a subset of classes projected a majority of OoD data points near InD data points, with only few OoD data points projected apart. This demonstrates the challenge of separating near-OoD data from InD data at point level.

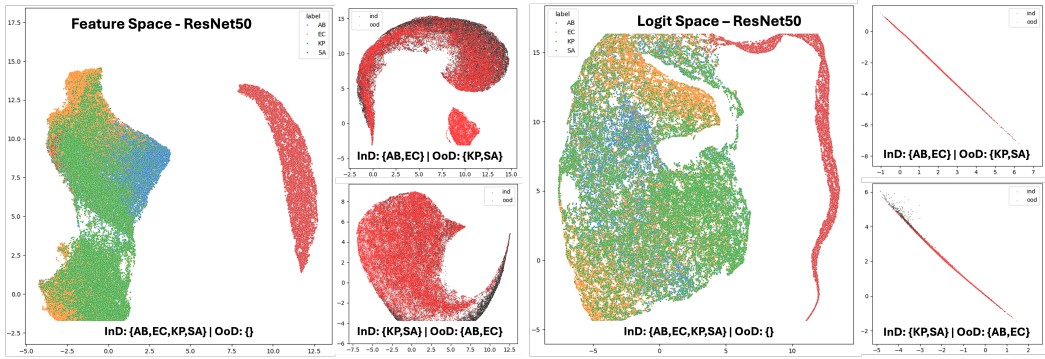

Figure 3: UMAP - Feature Space and Logit Space of **ResNet-50** model for AMRB Data. Higher InD-OoD separation is better. Data points are class-wise discriminative when the model is trained on all classes. However, this discriminative property is lost when the model is trained on a subset of classes. In this case, OoD classes fall into the same regions as InD classes, making point-wise OoD detection challenging on both spaces.

Fig. 4 visualizes UMAP projections of a ResNet-based classifier (ResNet-CAE) trained with a decoder and reconstruction loss for regularization. Here too, we observe similar latent space characteristics.

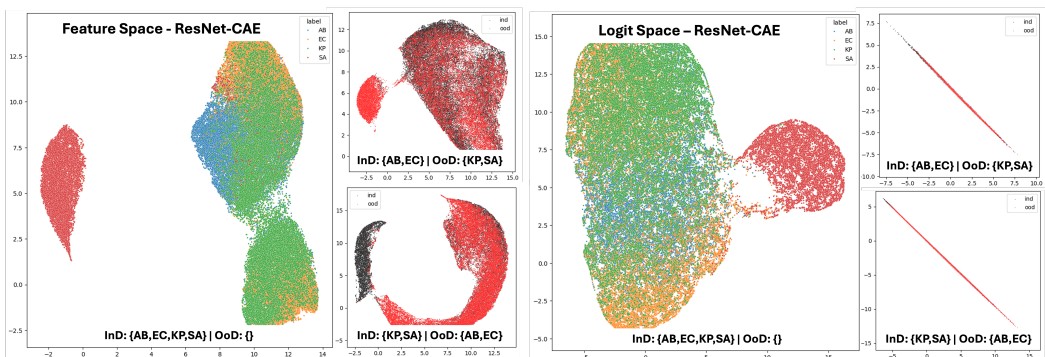

Figure 4: UMAP - Feature Space and Logit Space of **ResNet-CAE** model for AMRB Data. Higher InD-OoD separation is better.

In Fig. 5 and 6, we further illustrate point-level InD/OoD separability using three trained models – ResNet-50, ResNet-CAE, and ResNet-EDL, and two uncertainty metrics – $(1 - \mathrm{MSP})$ and $(1 - \sum \mathrm{logits})$. Fig. 5 plots the distributions of two point-wise OoD detection score functions for InD and OoD test data. Both score functions, $(1 - \mathrm{MSP})$ and $(1 - \sum \mathrm{logits})$, are based on predictive uncertainty. We visualize $(1 - \mathrm{MSP})$ from three models – ResNet-50 (classifier trained with cross-entropy loss), ResNet-CAE (classifier + autoencoder trained with cross-entropy + reconstruction loss) and ResNet-EDL (classifier trained with evidential loss, ensuring $\sum \mathrm{logits} \leq 1$). For the ResNet-EDL model, we also plot the distribution of $(1 - \sum \mathrm{logits})$. See the supplementary for the ResNet-EDL results. In most cases, we observe lower $(1 - \mathrm{MSP})$ scores for correctly classified InD data points than incorrectly classified InD data points. However, for OoD data points, instead of generating higher $(1 - \mathrm{MSP})$ scores as expected, we observe a more uniform distribution. Surprisingly, $(1 - \sum \mathrm{logits})$ generated seemingly worse scores than $(1 - \sum \mathrm{MSP})$. This demonstrates that point-wise OoD detection with score functions is also non-trivial.

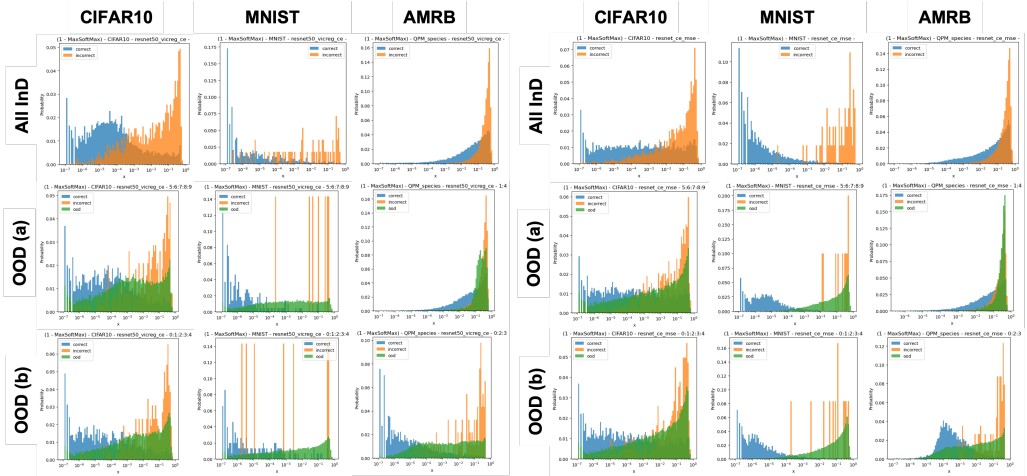

Figure 5: Uncertainty of Test Data. Higher InD-OoD separation is better. **Blue** - InD-Correct, **Orange** - InD-Incorrect, **Green** - OoD. **C1-C3:** $(1 - \mathrm{MSP})$ ResNet-50, **C4-C6:** $(1 - \mathrm{MSP})$ ResNet-CAE. Here, the uncertainty of OoD data points is distributed across the span of the metric, making it challenging to detect point-wise OoD using uncertainty.(**C** are columns from left to right)

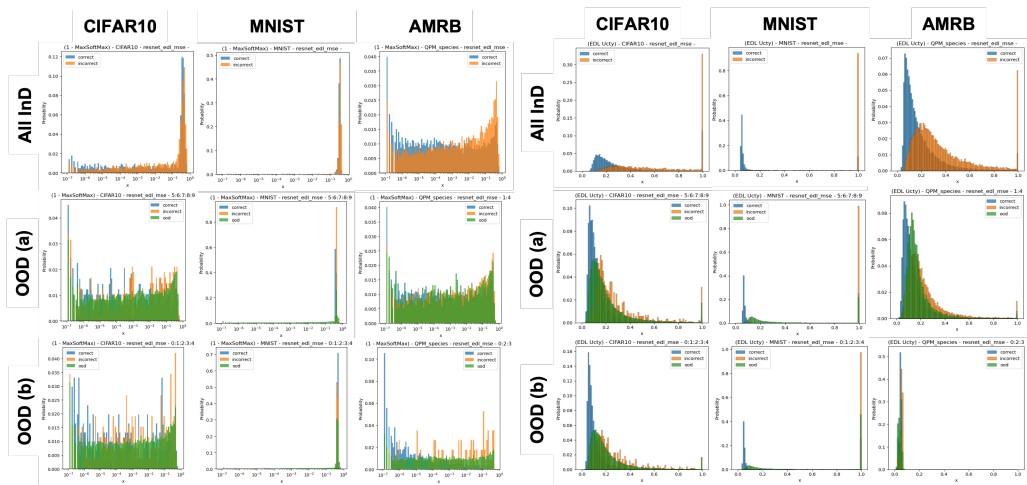

Figure 6: Uncertainty of Test Data. Higher InD-OoD separation is better. **Blue** - InD-Correct, **Orange** - InD-Incorrect, **Green** - OoD. **C1-C3:** $(1 - \text{MSP})$ ResNet-EDL, **C4-C6:** $(1 - \sum \text{logits})$ ResNet-EDL.

## A.7 ABLATION STUDIES

### A.7.1 ENSEMBLING OoD METRICS

In Fig. 7 , we AUC as the OoD detection performance metric and compare the performance of each OoD metric in isolation, to performance of using an ensemble of OoD metrics. We find that an ensemble of metrics consistently gives a high AUC. Our findings suggest that an ensemble of OoD metrics provides more consistent OoD detection performance across datasets and models.

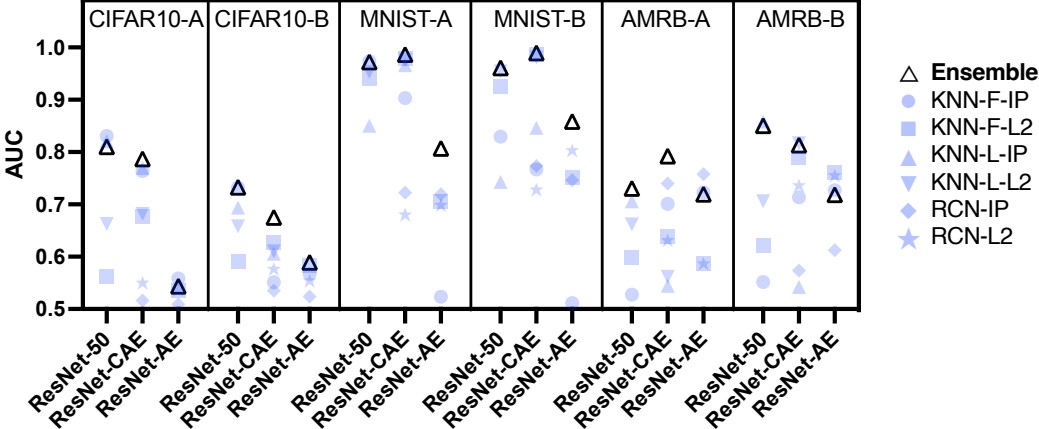

Figure 7: AUC of OoD Detection using Individual OoD Metrics vs Ensembling. Higher AUC is Better. Results are Averaged Across Comparisons.

### A.7.2 ENSEMBLING MODELS

In Fig. 8, we run hypothesis tests using combined OoD metrics from three model architectures (ResNet-50, ResNet-CAE, and ResNet-AE). Here, the metric vector used for the hypothesis test is the concatenation of the metric vectors of each model. where most comparisons were statistically significant except samples from the same homogeneous set. We did not, however, observe a substantial improvement in OoD detection AUC with ensembling.

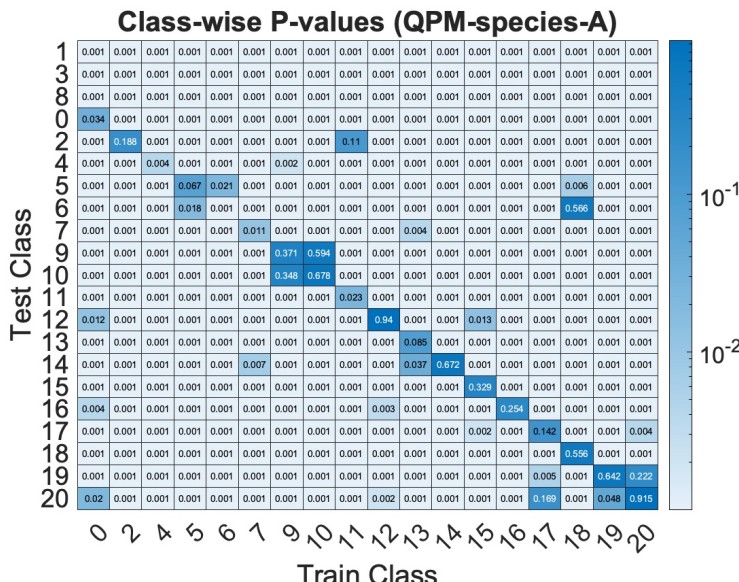

Figure 8: Observed p-values for AMRB dataset with MRPP statistic for a 3-model ensemble of OoD metrics. Each cell is a comparison between a test-set strain (across row) and a validation-set strain (across column). Permutations = 3000, Sample Size = 100.

### A.7.3 COMPARISON OF TEST STATISTICS AND MODELS

Fig. 9, Fig. 10 and Fig. 11 illustrate the observed statistic $\delta$ and its p-value $p$ for a set of tests, visualized in heatmaps form.

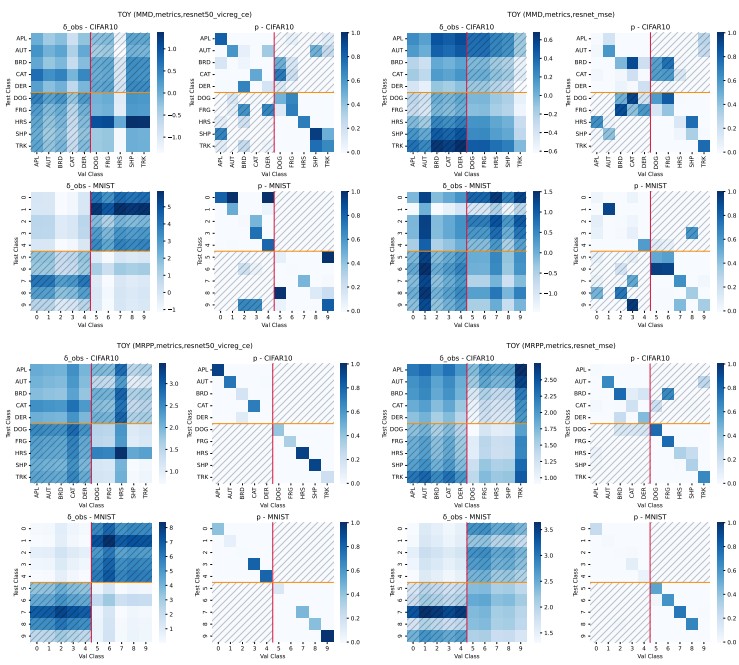

Figure 9: Observed test statistic (Odd Columns) and their p-values (Even Columns). **R1,R3:** CI-FAR10, **R2,R4:** MNIST, **R1–R2:** MD, **R3–R4:** MRPP, **C1–C2:** ResNet-50, **C3–C4:** ResNet-AE.

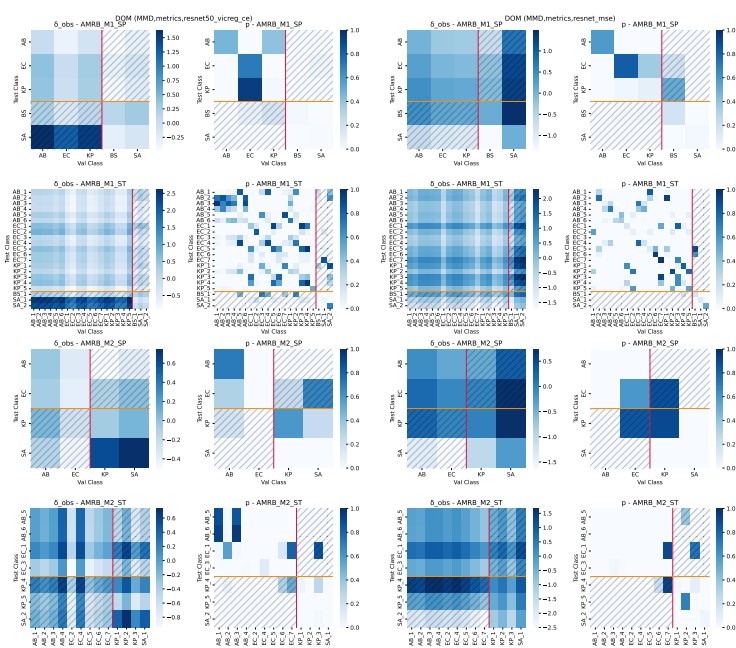

Figure 10: Observed test statistic (Odd Columns) and their p-values (Even Columns) for the ResNet-50 model. Statistic=MD **R1–R2:** Mutually non-exclusive val/test comparisons, **R3–R4:** Mutually exclusive val/test comparisons. **C1–C2:** ResNet-50, **C3–C4:** ResNet-AE.

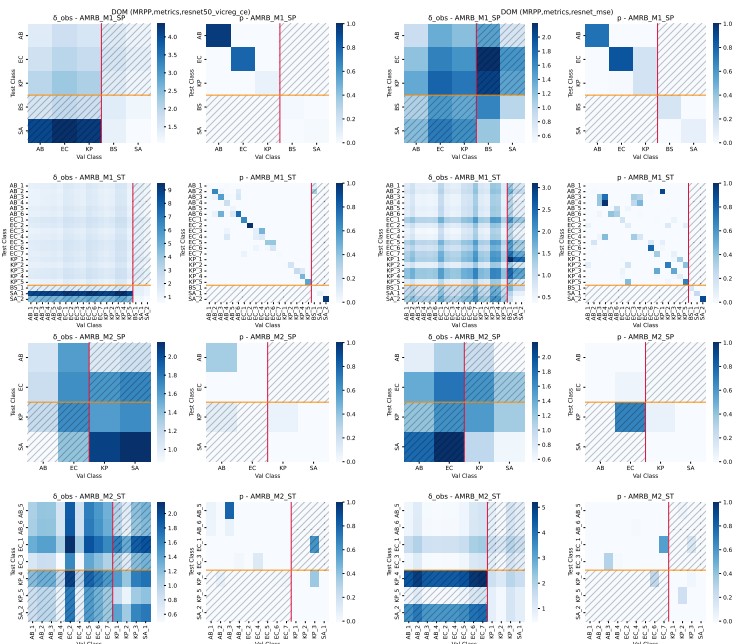

Figure 11: Observed test statistic (Odd Columns) and their p-values (Even Columns) for the ResNet-50 model. Statistic=MRPP. **R1–R2:** Mutually non-exclusive val/test comparisons, **R3–R4:** Mutually exclusive val/test comparisons, **C1–C2:** ResNet-50, **C3–C4:** ResNet-AE.

## A.8  HOoD Test Results: ResNet-50 Subsets

Fig. 12 reports the MRPP statistics and p-values of the individual HOoD tests reported in Table 1, prior to applying the decision function $D$.

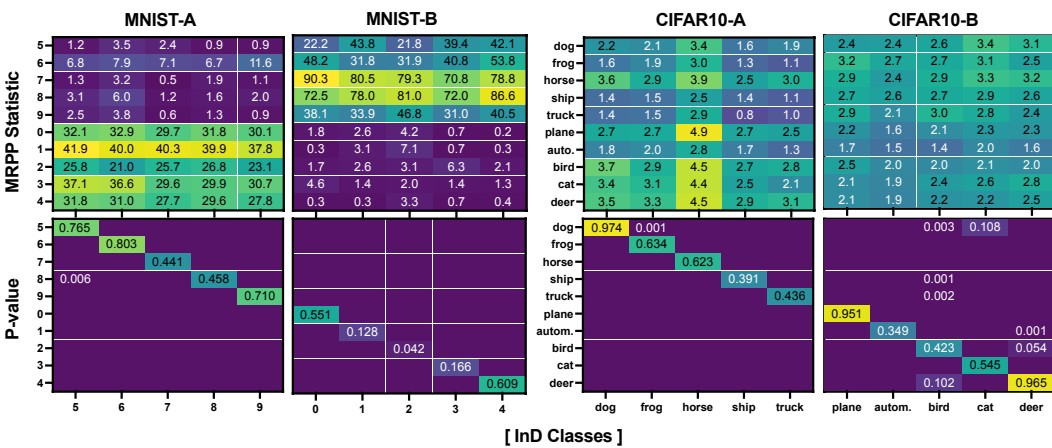

Figure 12: MRPP Statistics and P-values from HOoD tests performed using logit outputs of four ResNet-50 models on A/B splits of MNIST and CIFAR-10 datasets. Empty cells represent $p <$ 0.001. High $p$-values expected for HOoD tests on the diagonal (same class).

