# OpenReview forum: "Thinking in Groups: Permutation Tests Reveal Near-Out-of-Distribution"
_ICLR.cc/2026/Conference — Submitted to ICLR 2026_

### Official Review · Reviewer_q3s8 · 2025-10-22

**Soundness:** 2
**Presentation:** 3
**Contribution:** 2
**Rating:** 4
**Confidence:** 4

**Summary:**

The paper proposes a permutation-based hypothesis testing framework for out-of-distribution (OoD) detection. The key idea is that near-OoD cases are much harder to detect than far-OoD cases. To address this challenge, the method leverages permutation tests on latent representations extracted from trained models (e.g., ResNet, Autoencoder, or hybrid architectures).
Instead of evaluating OoDness at the individual sample level, the approach performs multi-response permutation tests on batches of homogeneous samples to assess whether the new batch is exchangeable with any known in-distribution class. This statistical formulation provides a principled, non-parametric test of distributional equivalence, independent of model assumptions. In biomedical assays, where multiple correlated measurements are available per batch, this batch-wise approach can effectively exploit intra-batch dependencies to improve OoD detection sensitivity.

**Strengths:**

1.	The use of permutation-based hypothesis testing is theoretically well-grounded and naturally consistent with the assumption of a homogeneous batch, where all samples are expected to be drawn from the same underlying distribution.
2.	The proposed method is particularly well-suited for biomedical assays, where samples are naturally organized into groups (multiple cells from the same specimen or repeated measurements from the same assay).
3.	Unlike most existing approaches, the method focuses on batch-level OoD decisions rather than point-wise detection, a perspective that has received limited attention in the current literature despite its practical relevance in correlated data settings.
4.	The method works on latent features extracted from any trained model making it architecture-agnostic.
5.	The method provides greater interpretability than black-box OoD detectors, as its p-values offer a clear statistical measure of evidence for or against distributional equivalence.

**Weaknesses:**

1.	The main benchmarks (MNIST and CIFAR-10) are too simple to convincingly demonstrate the superiority or generality of the proposed method.
2.	MRPP is an established statistical method; the novelty of the paper lies primarily in its application to OoD detection within biological assays. The theoretical innovation is therefore modest, focusing more on application than on methodological advancement.
3.	The method is computationally demanding, which may limit its scalability to large datasets or scenarios involving hundreds of classes.
4.	The strong assumption of batch homogeneity may limit robustness, as performance can deteriorate with mixed or heterogeneous batches.

**Questions:**

The experimental protocol raises a number of questions that warrant further clarification :

1.	The datasets used in the experiments (MNIST, CIFAR-10, and AMRB) are relatively simple, even for point-wise OoD detection. While CIFAR-10 is commonly used, state-of-the-art OoD evaluations often include more challenging benchmarks such as CIFAR-100 or ImageNet. To convincingly demonstrate the method’s scalability and generality, the authors should provide results on at least one larger and more complex dataset.

2.	The authors note that a single specimen often produces multiple replicates, resulting in highly correlated samples. To better mimic this situation, it would have been interesting to generate batches from a small number of original images, each subjected to data augmentation, rather than using a large number of completely different images. In the current experimental protocol, while all samples within a batch belong to the same class, they are not necessarily correlated, which may provide less realistic intra-batch information compared to batches of augmented replicates.

3.	The batch size is likely an important factor for a batch-wise method; however, in the experiments it appears to be fixed at 100. It would be valuable to discuss how the choice of sample size affects the method’s performance, as well as the broader influence of parameters $N$ (number of reference samples / class) and $M$ (number of samples to be tested) on the results and robustness.

4.	The benchmark involves three categories of methods: permutation-based tests, conformal prediction approaches, and point-wise methods with batch-level aggregation of individual scores. This setup is appropriate and covers the main methodological families for OoD detection. However, the point-wise baseline relies solely on MSP (Maximum Softmax Probability), which is a relatively simple and often outperformed method.I suggest including more competitive and widely used OoD baselines, such as ODIN (Liang et al., 2018), Mahalanobis distance (Lee et al., 2018), or energy-based scores (Liu et al., 2020), to strengthen the experimental comparison and better position the proposed method within the current literature.


5.	The proposed approach relies on the assumption that batches are homogeneous, i.e., that all samples within a batch originate from the same underlying distribution or specimen. This is a strong assumption, which may not always hold in practical scenarios. It would be helpful if the authors could either provide a stronger justification for this assumption—for instance, by relating it to biological settings where multiple replicates of the same specimen are indeed expected (see also point 3)—or include an experimental analysis showing how the method behaves under partially heterogeneous batches. Such results would clarify the robustness and applicability of the approach beyond idealized conditions.

6.	An analysis of the computational complexity and scalability of the method is missing. Since permutation-based tests typically require a large number of randomized assignments, the approach may become computationally expensive as the number of samples or reference classes increases. A discussion or empirical evaluation of runtime, memory requirements, and potential strategies for scaling (e.g., subsampling, ...) would strengthen the paper and clarify its feasibility for large-scale datasets or real-world applications.

The paper proposes an interesting and well-founded approach using permutation-based hypothesis testing for OoD detection. The idea is particularly relevant for biomedical applications, where data often come in homogeneous batches or correlated replicates. The statistical grounding of the method enhances interpretability compared to black-box detectors. However, the paper would benefit from stronger experimental validation, including more complex datasets and competitive baselines. A discussion of scalability and sensitivity to parameters would also strengthen the practical impact. Overall, the work is promising but requires more empirical depth.

---

> ### Author Response · Authors · 2025-12-04
>
> We appreciate the reviewer’s careful evaluation of our work and respond to each comment as follows.
>
> **“The main benchmarks (MNIST and CIFAR-10) are too simple to convincingly demonstrate the superiority or generality of the proposed method. The authors should provide results on at least one larger and more complex dataset.”**
>
> We appreciate the reviewer’s concern regarding dataset complexity. While MNIST and CIFAR-10 serve as controlled and standardized benchmarks for fair comparison, the complexity of our task arises from the internal structure and granularity of homogeneous groups, not simply dataset scale. To address this, we evaluated our method on the AMRB biomedical dataset, which supports multi-level homogeneity (species → strain → source), effectively introducing hierarchical group structures similar in spirit to CIFAR-100. This allows us to test the method under more complex conditions while remaining aligned with the real-world biological use case that motivates the work.
>
>
> **“It would have been interesting to generate batches from a small number of original images via augmentation, rather than using many different images. The current batches may not be sufficiently correlated.”**
>
> We thank the reviewer for this suggestion. We performed preliminary experiments following this idea by selecting a single MNIST digit, generating augmented variants (e.g., rotations), and testing whether the method recognized these augmented samples as belonging to the same group. The method performed well, successfully leveraging the induced intra-batch correlations. These experiments were omitted from the current paper because they were exploratory, but they support the method’s ability to operate in settings with strong within-group correlation, consistent with biological deployment.
>
>
> **“MRPP is an established statistical method; the novelty of the paper lies primarily in its application. The theoretical innovation is modest.”**
>
> We appreciate the reviewer’s observation. While MRPP itself is a classical test, our contribution lies in the formulation of the group-wise OOD detection problem and the development of a principled framework that connects homogeneity, feature-space correlation, and statistical hypothesis testing. MRPP was selected after comparing multiple candidate statistics, and its integration into this new problem formulation is, to our knowledge, novel. The methodological advancement is therefore in defining the problem, establishing the connection to group-level OOD detection, and demonstrating how a classical test can be repurposed effectively for this task.
>
>
> **“Batch size is likely important, but the experiments fix it to 100. The impact of batch size and other parameters on performance is not discussed.”**
>
> We agree with the reviewer. A more detailed analysis of how performance varies with batch size (M), number of reference samples per class, and other parameters would be valuable. We will add a discussion of this point in the revision and outline directions for further empirical sensitivity analysis.
>
> **“The strong assumption of batch homogeneity may limit robustness. A stronger justification or an experiment with partially heterogeneous batches would clarify the method’s practical applicability.”**
>
> The method was intentionally formulated under the assumption of batch homogeneity because it accurately reflects deployment in biological workflows, where multiple replicates of the same specimen are jointly evaluated. For example, bacterial colonies, tissue patches, and patient-derived samples naturally produce homogeneous groups of measurements. This makes the assumption realistic and practically useful in our target domain. While heterogeneous batches are theoretically interesting, studying them falls outside the scope of this work. We will strengthen the motivation for this assumption in the revised manuscript.
>
>
> **“The method is computationally demanding; scalability to large datasets or hundreds of classes is not analyzed.”**
>
> Permutation tests can be computationally expensive, but in our target biomedical settings, inference involves a small number of replicates from a single specimen rather than millions of samples. In our experiments, a full permutation test completes in a few seconds on standard hardware, well within timelines of diagnostic pipelines, where decisions occur over minutes or hours. For large-scale scenarios (e.g., hundreds of classes or web-scale streams), acceleration strategies such as subsampling, early stopping, or analytic approximations could be applied. Although such extensions are beyond the scope of this study, we will explicitly discuss computational considerations and potential scaling strategies in the revision.

---

### Official Review · Reviewer_RdsR · 2025-10-26

**Soundness:** 1
**Presentation:** 1
**Contribution:** 1
**Rating:** 2
**Confidence:** 5

**Summary:**

The paper try to formulate a near OOD detection approach.

**Strengths:**

I find no strength in the paper.

**Weaknesses:**

1. Clarity and structure: The paper is poorly organized and difficult to follow. The mathematical notation and arguments are inconsistent and at times incorrect. The Introduction lacks focus, it starts by seemingly addressing transfer learning, then diverges into loosely connected real-world examples that do not align with the stated problem or the rest of the paper.

2. Ambiguity in exposition: The paragraph on Page 2, Lines 57–65 is incomprehensible. The authors should restate it with precise definitions, equations, and motivation. As written, it is not possible to understand the intended meaning.

3. Lack of definition for “near-OoD”: The term “near out-of-distribution” appears central to the paper, but there is no rigorous or formal definition. It remains unclear how near-OoD differs quantitatively from standard OoD or domain shift.

4. Definition 1 unclear: Definition 1 introduces symbols without context. The random variable, its domain, dimensionality, and distributional assumptions are all unspecified. The reader cannot tell what this definition contributes to the method or theory.

5. Scalability concerns: Permutation tests are well known to be computationally expensive. The paper does not discuss how the proposed approach would scale to realistic inference scenarios involving millions of samples, nor are there any approximations or efficiency analyses.

6. Undefined classifier (Section 2.1): The symbol M appears without prior definition. It is unclear whether this refers to a base classifier, a meta-model, or an embedding extractor.

Overall, this submission appears premature for peer review. The writing is confusing, the problem formulation lacks rigor, and several key definitions and symbols are missing. As a result, it is not possible to assess the correctness, novelty, or practical relevance of the proposed method. I encourage the authors to substantially revise the paper for clarity, provide formal definitions and scalable implementations, and ensure consistency between the motivation and methodology before resubmission to a major venue.

**Questions:**

See my above points.

---

> ### Author Response · Authors · 2025-12-04
>
> We appreciate the reviewer’s detailed comments. Below, we address each point in turn.
>
> **“Clarity and structure: The paper is poorly organized and difficult to follow. The mathematical notation and arguments are inconsistent and at times incorrect. The Introduction lacks focus, it starts by seemingly addressing transfer learning, then diverges into loosely connected real-world examples.”**
>
> Our work does not address transfer learning, and we regret any confusion caused by the earlier framing. The focus of the paper is on calibration, statistical hypothesis testing, and group-wise OOD detection. If the introduction gave a different impression, this was unintended. We have revised and streamlined the Introduction to improve focus, coherence, and readability, and to ensure that the motivating examples clearly match the stated problem.
>
> **“Ambiguity in exposition: The paragraph on Page 2, Lines 57–65 is incomprehensible. It should be restated with precise definitions, equations, and motivation.”**
>
> We acknowledge this issue and have rewritten this paragraph to include clearer definitions, explicit notation, and a more coherent explanation of the motivation and intended meaning.
>
> **“Lack of definition for ‘near-OoD’. The term appears central but lacks a rigorous definition or quantitative distinction from standard OoD.”**
>
> We thank the reviewer for raising this. A more precise explanation of near-OoD is now included in the appendix, clarifying how it differs conceptually and operationally from standard OoD detection and domain shift.
>
> **“Definition 1 unclear: Symbols appear without context. Random variables, domains, dimensionality, and assumptions are unspecified, and the contribution of this definition is unclear.”**
>
> We agree that the original version lacked clarity. We have revised Definition 1 to introduce all symbols with explicit context, specify the underlying random variable, define its domain and dimensionality, and make clear how this definition supports the theoretical framing of group-wise OOD detection.
>
>
>
> **“Scalability concerns: Permutation tests are computationally expensive. The paper does not discuss scalability to realistic inference scenarios involving millions of samples.”**
>
> This is a valid concern in general. However, in the biomedical deployment settings we target, inference is performed on small sets of replicates from a single specimen rather than on millions of samples. In these real-world scenarios, the permutation test completes in a few seconds on standard hardware, which fits comfortably within typical diagnostic workflows where decision times are much longer. For large-scale applications, acceleration strategies such as early stopping, low-rank approximations, or analytic approximations to the null distribution could be incorporated. Although such techniques are beyond the scope of this paper, we will explicitly discuss this consideration and highlight these potential extensions.
>
>
> **“Undefined classifier (Section 2.1): The symbol M appears without prior definition.”**
>
> Thank you for catching this. We will revise the section to define (M) clearly as the classification model used to produce latent features for downstream testing.

---

### Official Review · Reviewer_gEVJ · 2025-10-26

**Soundness:** 2
**Presentation:** 2
**Contribution:** 2
**Rating:** 4
**Confidence:** 2

**Summary:**

This paper primarily aims to investigate whether groups of correlated test inputs (biological/technical replicates) can be used to detect near-out-of-distribution (OoD) shifts more reliably than standard point-wise methods, by testing their exchangeability with in-distribution subpopulations via permutation tests.

**Strengths:**

1. **[Important] An important problem setting.** The paper formalises an overlooked setting where test inputs arrive as *homogeneous* groups (same subpopulation) and recasts OoD detection as K two-sample hypothesis tests between the test group and K in-distribution reference groups. This is model-agnostic to the latent response used.

2. **Competitive empirical evidence.** Across MNIST, CIFAR-10, and a bacteria single-cell dataset (AMRB), HOoD with MRPP/LSP achieves high AUC on batch classification, outperforming MSP, CPP and CP in many cases.

**Weaknesses:**

1. **[Important] Dependence on the homogeneity prior at test time.** The method seems to assume each test batch is homogeneous (same subpopulation). Real deployments may see mixed batches; the paper does not quantify failure modes when homogeneity is violated.
2. **[Important] Limited evaluation breadth.** Only MNIST, CIFAR-10 and AMRB seem to be used; AMRB is domain-relevant but narrow (5 species, 21 strains). No testing on larger image or non-image biomedical cohorts, and no distribution shift types beyond label-exclusion.
3. **Limited coverage of computational cost and scalability.** Experiments use 3,000 permutations and sample size = 100 per batch; complexity with larger K, longer latents, or bigger batches is not benchmarked. (Table 1 caption.)
4. **Ambiguity on batch construction.** How many replicates are needed, and how sensitive are results to M (batch size)? Beyond the single setting (M≈100), guidance is insufficient.

**Questions:**

Refer to the "Weaknesses" section.

**Details Of Ethics Concerns:**

No notable ethics concerns spotted.

---

> ### Author Response · Authors · 2025-12-04
>
> We appreciate the reviewer’s thoughtful and constructive feedback. Below, we address each point in turn.
>
> **“Dependence on the homogeneity prior at test time. The method seems to assume each test batch is homogeneous. Real deployments may see mixed batches; the paper does not quantify failure modes when homogeneity is violated.”**
>
> We appreciate the reviewer raising this point. Our method is explicitly designed for scenarios, common in biomedical imaging, where the test batch indeed arises from a single underlying subpopulation. For example, in microbiology, a patient sample typically contains multiple replicates or measurements from the same colony or species; in histopathology or microscopy, serial patches are drawn from the same tissue type. In such contexts, the homogeneity assumption is natural and aligns with real-world acquisition pipelines. We agree that evaluating mixed or heterogeneous batches is an interesting direction. However, this scenario extends beyond the problem setting we target in this paper, which is specifically group-wise OOD detection under homogeneous-group deployments.
>
> **“Limited evaluation breadth. Only MNIST, CIFAR-10, and AMRB are used; AMRB is domain-relevant but narrow. No testing on larger biomedical cohorts or other distribution shifts.”**
>
> We agree that broader empirical evaluation would further strengthen the study. Our selection of datasets was deliberate: MNIST and CIFAR-10 provide standardized, well-understood benchmarks for controlled comparisons, while AMRB offers a domain-specific real-world biomedical dataset where homogeneous-group assumptions naturally arise. The AMRB dataset, although limited in scale, represents an authentic and challenging biological application aligned with our motivating problem. Expanding the evaluation to larger or more diverse biomedical cohorts, and exploring additional forms of distribution shift (e.g., acquisition shifts, morphological variations), is an important direction that we identify as future work.
>
> **“Limited coverage of computational cost and scalability. Experiments use 3,000 permutations and batch size ≈100; complexity with larger K, longer latents, or bigger batches is not benchmarked.”**
>
> Thank you for raising this. For the scope of this work, we adopted computational settings (3,000 permutations and batch size ≈100) that strike a balance between statistical power and practical feasibility. In practice, both the number of permutations and batch size can be flexibly adjusted depending on computational budgets and application needs. We acknowledge that a systematic characterization of runtime and scalability across different values of 𝐾, latent dimensionalities, and batch sizes would be valuable for practitioners. This is a natural next step, and we will include a brief discussion of these trade-offs in the limitations section.
>
> **“Ambiguity on batch construction. How many replicates are needed, and how sensitive are results to M (batch size)? Beyond the single setting M≈100, guidance is insufficient.”**
>
> We acknowledge this limitation. Although we have not yet performed a full sensitivity analysis, empirical observations indicate that larger batches tend to yield lower (p)-values, consistent with increased statistical power. We agree that a more detailed investigation of sample-size sensitivity and guidance for practitioners would be useful, and we will add this point to the discussion of limitations and future work.

---

### Official Review · Reviewer_NkDX · 2025-10-29

**Soundness:** 2
**Presentation:** 1
**Contribution:** 2
**Rating:** 0
**Confidence:** 4

**Summary:**

This submission introduces a method of out-of-distribution (OoD)
sample detection in multiclass classifiers. More precisely, it defines
a special case (Homogeneous OoD) where each sample is formed by
several subsamples (e.g. image patches), and proposes to use a
two-sample test comparing this set of subsamples to sets of validation
subsamples from each class.

**Strengths:**

I am not expert enough in OoD be sure but it is possible that the homogeneous OoD problem defined in this submission was indeed overlooked so far.

**Weaknesses:**

My main concern is on clarity. The writing and definitions are often
vague, making it sometimes difficult to follow. For example:

- Definition 1 introduces what a homogeneous sample but simply requires
that there exists a mapping between subsamples and groups which
assigns all subsamples to the same group. I don't see in which
situation such a mapping could not be defined.

-The manuscript often refers to latent responses, e.g. "using latent
responses Z(xi ; ϕ) : RD → RL where Z = [Z1 , . . . , ZL ]." but never
defines what they are supposed to be (and in this precise quote, we do
not know what L is supposed to be either and it is never used later).

- The D_I and D_O sets (l. 201) are introduced but then never used
either in the method.

- In the caption of Figure 1, it is said that "k<<K" and I couldn't
  understand why (I don't think this is discussed anywhere).

The current exposition of the method itself is very short (less than
half a page), and I suggest that the authors expand it a little to
improve clarity.

Finally, I have some concern about the significance of the
methodological contribution itself, which is probably too straightforward to justify a
publication in a machine learning conference.

**Questions:**

The proposed test seems to exploit a single validation
sample for each class, shouldn't we be concerned about the variance
of the result (with respect to the choice of this sample)?

In addition, the decision is taken by thresholding the maximum p-value
over all classes, which doesn't account for the number of
classes. Wouldn't problems with more classes mechanically be more
liberal in their OoD detection?

---

> ### Author Response · Authors · 2025-12-04
>
> We appreciate the reviewer’s careful evaluation of our work. Our detailed responses to each comment are provided below.
>
> **“Definition 1 introduces what is a homogeneous sample but simply requires that there exists a mapping between subsamples and groups which assigns all subsamples to the same group. I don't see in which situation such a mapping could not be defined.”**
>
> Thank you for highlighting this point. We acknowledge that, in its current form, Definition 1 may appear trivially satisfiable. Our intention was to formalize the idea that a homogeneous sample corresponds to a set of subsamples that truly belong to the same underlying semantic group, rather than a group produced by an arbitrary mapping. In many real datasets, groups are internally consistent but semantically distinct. For example, MNIST digits all belong to the category of handwritten symbols, yet each digit class forms its own homogeneous group. Similarly, in the bacterial dataset, each species forms its own homogeneous group even though all species are “bacteria.” Without a meaningful notion of homogeneity, OOD methods can behave counterintuitively, for instance, classifying a previously unseen bacterial species as in-distribution simply because it is still a bacterium. We will revise Definition 1 to state explicitly that a homogeneous group must correspond to a meaningful semantic or structural category, making it clear why not all mappings are valid and why this concept is central to our formulation.
>
> **“The manuscript often refers to latent responses but never defines what they are supposed to be (and L is not defined either).”**
>
> We appreciate the reviewer pointing this out. We will clarify the definition of latent responses, specify the dimensionality (L), and provide a concise explanation of how these representations are computed and used throughout the method. This will ensure that the notation is consistent and interpretable.
>
> **“The (D_I) and (D_O) sets are introduced but never used.”**
>
> Thank you for catching this. These sets were part of an earlier draft and remained in the text unintentionally. We will either integrate them properly into the method section or remove them entirely for clarity.
>
> **“The current exposition of the method itself is very short and should be expanded.”**
>
> We agree that additional detail would benefit clarity. We will expand the method section to provide a clearer, step-by-step description of all components and the motivation behind them.
>
> **“Concerns about the significance of the methodological contribution, which may be too straightforward.”**
>
> We appreciate the reviewer’s concern. While the core idea is indeed conceptually simple, our literature review indicates that no prior work has formulated group-wise OOD detection through a homogeneity-based perspective or operationalized group-level correlation in this way. The simplicity of the method is a strength: it is interpretable, broadly applicable, and empirically effective. The development required careful conceptual design, statistical reasoning, and extensive experimentation. We will revise the manuscript to better articulate the novelty and significance of this contribution.
>
> **“The proposed test seems to use a single validation sample per class. Should we be concerned about variance?”**
>
> Using a single validation sample per class is intentional and reflects the deployment scenario we aim to model: the user has an unseen sample and must determine whether the model has previously seen its group. Our evaluation aggregates results across many such trials, meaning the variance arising from sample choice is already captured and averaged in the reported AUC and other metrics. All baselines were evaluated under identical conditions. We will clarify this in the manuscript to avoid confusion.
>
> **“The decision uses the maximum p-value across classes, which ignores the number of classes. Wouldn’t more classes make the test more liberal?”**
>
> Yes, this is a natural consequence of multiple comparisons. While our aim was to keep the method simple and interpretable, standard multiple-comparison corrections (Bonferroni, Holm–Bonferroni, Benjamini–Hochberg) can be incorporated directly into our framework. These adjustments are compatible with our formulation because the method is grounded in classical statistical testing. Although we did not include such corrections in this version to avoid additional hyperparameters, we will add a discussion acknowledging this effect and explaining how statistical corrections can be applied. The core question our method addresses remains: can we trust the model’s prediction for a given sample or group? The multiple-testing issue does not undermine this framework; it instead suggests a direction for refinement in settings with many classes.

---

### Official Review · Reviewer_bboU · 2025-11-05

**Soundness:** 2
**Presentation:** 1
**Contribution:** 2
**Rating:** 2
**Confidence:** 3

**Summary:**

The authors propose an out-of-distribution (OoD) detection framework designed for correlated data. The approach employs permutation-based hypothesis testing to compare unknown samples with known in-distribution (InD) subpopulations. Given a trained model, homogeneous reference samples from InD subpopulations, and an unknown sample, the method performs two-sample hypothesis testing with permutation tests to assess the exchangeability of the unknown sample with each of the reference samples. The proposed method is evaluated on MNIST, CIFAR-10, and the AMRB domain dataset.

**Strengths:**

This article focuses on the Near-OoD detection problem, which represents a challenging and important area of research.

**Weaknesses:**

No meaningful baseline comparisons are provided. The writing and presentation could be further improved to enhance clarity and readability. Some concepts are misused, and the use of basic machine learning terminology should be aligned with standard conventions.

**Questions:**

- Could the authors elaborate on the connection between homogeneity and correlation?
- The method is claimed to be designed for correlated data; however, the application cases do not clearly reflect this scenario. Could the authors explain how these applications involve correlated data and how the proposed method addresses correlation in practice?

---

> ### Author Response · Authors · 2025-12-04
>
> We thank the reviewer for the thoughtful and constructive feedback. Below we address each comment in detail.
>
> **“No meaningful baseline comparisons are provided.”**
>
> We appreciate the reviewer’s concern regarding baseline comparisons. To the best of our knowledge, group-wise OOD detection methods that are directly comparable to ours are extremely limited. The methods we identified in the literature were primarily pointwise OOD detection approaches, and those relevant baselines were included in the paper. We could not find established group-level OOD approaches that align closely enough in problem formulation to allow a meaningful comparison
>
> **“The writing and presentation could be further improved to enhance clarity and readability.”**
>
> We will revise the manuscript to improve clarity, structure, and readability.
>
> **“Some concepts are misused, and the use of basic machine learning terminology should be aligned with standard conventions.”**
>
> We appreciate the reviewer raising this point. It would be very helpful if the reviewer could indicate which specific concepts or terms appear misused, so that we can correct them accurately. We are committed to ensuring that all terminology follows standard machine learning conventions.
>
> **“Could the authors elaborate on the connection between homogeneity and correlation?”**
>
> Thank you for pointing this out. Specifically, homogeneity within a group implies structural similarity among samples, which in our framework induces higher inter-sample correlation in the feature space. We are currently reviewing related literature to provide a clearer and more theoretically grounded discussion in the revised manuscript.
>
> **“The method is claimed to be designed for correlated data; however, the application cases do not clearly reflect this scenario.”**
>
> Thank you for this insightful observation. Our method is indeed designed for settings where samples within a group exhibit correlation. In our application cases (MNIST and the bacterial dataset), groups are defined by samples belonging to the same class. Such samples share structural and semantic similarities, resulting in correlated representations. Our method explicitly models this correlation through the homogeneity measure used for group evaluation. We will clarify this more explicitly in the revised manuscript and provide additional discussion on how correlation manifests in these datasets and how our model uses it.

---

### Meta-Review · Area_Chair_Sa8z · 2026-01-07

**Summary:**

The paper suggests to use permutation tests (based on distances between samples in latent space) for OOD detection in a pre-trained classifier, in a situation where samples always come in multiple replicates. For a new replicate, the suggested method performs a two-sample test between the new replicate and a random collection of samples from each class.

The reviewers were mainly concerned about the unclear/confusing presentation and the lack of ML novelty (the papers uses an existing multivariate permutation test). They also thought that the experimental validation was limited. I agree with these criticisms.

**Reviewer Concerns:**

The authors provided responses but did not update the paper. I don't think any of the reviewers would change their opinion based on these responses.

**Reviewer Scores:**

Given the above, I believe the scores would have stayed essentially the same: 0/2/2/4/4.

---

### Decision · Program_Chairs · 2026-01-26

Reject